# E-Cigarette Aerosol Exposure Favors the Growth and Colonization of Oral *Streptococcus mutans* Compared to Commensal Streptococci

Alma Catala-Valentin,[a] Joshua N. Bernard,[a] Matthew Caldwell,[a] Jessica Maxson,[a] ⓘ Sean D. Moore,[a] ⓘ Claudia D. Andl[a]

[a]Burnett School of Biomedical Sciences, College of Medicine, University of Central Florida, Orlando, Florida, USA

Alma Catala-Valentin and Joshua N. Bernard contributed equally to this article. Author order was determined by seniority.

**ABSTRACT** E-cigarettes (e-cigs) have drastically increased in popularity during the last decade, especially among teenagers. While recent studies have started to explore the effect of e-cigs in the oral cavity, little is known about their effects on the oral microbiota and how they could affect oral health and potentially lead to disease, including periodontitis and head and neck cancers. To explore the impact of e-cigs on oral bacteria, we selected members of the genus *Streptococcus*, which are abundant in the oral cavity. We exposed the commensals *Streptococcus sanguinis* and *Streptococcus gordonii* and the opportunistic pathogen *Streptococcus mutans*, best known for causing dental caries, to e-liquids and e-cig aerosols with and without nicotine and with and without menthol flavoring and measured changes in growth patterns and biofilm formation. Our results demonstrate that e-cig aerosols hindered the growth of *S. sanguinis* and *S. gordonii*, while they did not affect the growth of *S. mutans*. We also show that e-cig aerosols significantly increased biofilm formation by *S. mutans* but did not affect the biofilm formation of the two commensals. We found that *S. mutans* exhibits higher hydrophobicity and coaggregation abilities along with higher attachment to OKF6 cells than *S. sanguinis* and *S. gordonii*. Therefore, our data suggest that e-cig aerosols have the potential to dysregulate oral bacterial homeostasis by suppressing the growth of commensals while enhancing the biofilm formation of the opportunistic pathogen *S. mutans*. This study highlights the importance of understanding the consequences of e-cig aerosol exposure on selected commensals and pathogenic species. Future studies modeling more complex communities will provide more insight into how e-cig aerosols and vaping affect the oral microbiota.

**IMPORTANCE** Our study shows that e-cigarette aerosol exposure of selected bacteria known to be residents of the oral cavity hinders the growth of two streptococcal commensals while enhancing biofilm formation, hydrophobicity, and attachment for the pathogen *S. mutans*. These results indicate that e-cigarette vaping could open a niche for opportunistic bacteria such as *S. mutans* to colonize the oral cavity and affect oral health.

**KEYWORDS** e-cigarette, *S. mutans*, streptococci, e-cigarette aerosols, oral streptococci, oral health

Address correspondence to Claudia D. Andl, claudia.andl@ucf.edu.

The authors declare no conflict of interest.

E-cigarette (e-cig) use has increased drastically in the last decade, with 4.7% of middle-school students and 19.6% of high-school students currently using an e-cig device (1–3). The base components of e-liquids are propylene glycol (PG) and vegetable glycerin (VG), used in various ratios. Additional components include nicotine and flavorings that can be added by the e-cig user (4). Among adolescents who vape any

type of flavored e-cigarettes, the most commonly used flavors are fruit (73.1%; 1.83 million users), mint (55.8%; 1.39 million), and menthol (37.0%; 920,000) according to the U.S. Department of Health and Human Services (2). These e-liquids are heated into an aerosol using an electronic nicotine delivery system (ENDS) and are orally inhaled by the user (so-called "vaping"). While e-cigs are advertised as a safer alternative to conventional cigarettes (5), there is a lack of long-term epidemiological data available to support this claim (4). It has been reported that e-cigs generate reactive oxidative species (ROS) and other carcinogenic components, such as formaldehyde, nitrosamines, and toxic carbonyls (6–8), similar to those present in traditional cigarette smoke condensate (9–11). These e-cig metabolites have been shown to induce DNA damage, thereby enhancing mutational susceptibility and tumorigenic transformation (12, 13).

Oral bacteria and oral epithelial cells are directly exposed to e-cig aerosols during a vaping session. E-cig vaping has been found to affect oral health through the induction of inflammatory-cytokine release and oxidative damage (14). A metagenomic study showed that e-cig aerosol exposure increased the abundance of bacterial genes encoding quorum sensing, biofilm formation, stress response, and virulence factors (15). However, current data regarding the effects of e-cigs on the oral microbiome are limited and controversial. While one report showed that e-cigs did not affect oral microbiome diversity (16), others agreed on changes in the oral microbial diversity and differences in the relative abundance of several bacterial taxa, including *Porphyromonas*, *Veillonella* (1), and *Haemophilus*, in e-cig users compared to healthy individuals (17).

The oral microbiome is dominated by oral commensal *Streptococcus* spp. (80%) (18, 19), leading to our interest in investigating the effect of e-cig aerosols in the modification of these oral resident bacteria. At homeostasis, *Streptococcus sanguinis* and *S. gordonii* dominate the niche. Two recent studies showed that e-liquid and e-cig aerosols did not cause changes in the exponential growth phase of *Streptococcus* commensals, including *S. gordonii*, *S. mitis*, and *S. oralis* (20, 21). In contrast, another recent study demonstrated that flavored e-cig aerosols hindered the exponential growth of *S. gordonii*, *S. intermedius*, *S. mitis*, and *S. oralis* more strongly than unflavored e-cig aerosols (22). *Streptococcus mutans* is an opportunistic pathogen known for its cariogenic role in the oral cavity (18) and its association with severe periodontitis (23, 24). Typically, *S. mutans* is present in the oral cavity as part of a mature dental biofilm. However, it can become dominant over other species due to environmental changes, including density, nutritional availability, and pH, causing dysregulation in oral bacterial homeostasis (25). Kreth et al. studied competition and coexistence between different species occupying the same ecological niche using *S. mutans* and *S. sanguinis* as a model (25). Based on their data, occupation of a niche by one species prevents colonization by the other, yet simultaneous colonization by both species results in coexistence (25). Currently, there are few data available regarding the effect of e-cig aerosols on the colonization of the oral cavity by *Streptococcus* spp. To better understand how e-cigs affect these oral interspecies interactions, we focused on the effect e-cig aerosols had on growth patterns and biofilm formation of two known early colonizers, the commensals *S. sanguinis* and *S. gordonii*, as well as the opportunistic pathogen and recognized cariogenic bacterium *S. mutans*. Our results show that e-cig aerosols reduce the biomass accumulation of *S. sanguinis* and *S. gordonii* without affecting the biomass accumulation of *S. mutans*. E-cig aerosols promoted *S. mutans* biofilm formation but had no effect on *S. sanguinis* and *S. gordonii*. When we assessed bacterial hydrophobicity and growth rate, we showed dominance of *S. mutans* over the commensal family members, demonstrating the potential role of e-cig vaping to shift the bacterial population by altering growth and colonization of every species differently.

## RESULTS

**E-cigarette aerosols suppress the growth of *S. sanguinis* and *S. gordonii* but not *S. mutans*.** In this study, we explored the effect of the common propylene glycol-vegetable glycerin (PG/VG; 50/50) e-cig base without any additives and with 3 mg/mL nicotine.

Since menthol is one of the most popular flavorings, we also included treatments with nicotine-free menthol flavoring and menthol containing 3 mg/mL nicotine. These e-liquids were heated into aerosols using an electronic nicotine delivery system (ENDS). For this study, we focused on closely related commensals from the *Streptococcus sanguinis* group, *S. sanguinis* and *S. gordonii*, which are reported to be early colonizers and able to adhere to all oral cavity surfaces (18). We also selected *S. mutans*, which is a known opportunistic pathogen that under certain condition can become dominant over other species and cause caries and periodontal disease (25). To assess competition between *S. mutans* and *S. gordonii* or *S. sanguinis*, we cultured each of the commensals, adding *S. mutans* in an equal volume for 24 h. *S. mutans* growth increased by 32-fold ($P < 0.01$) by the end of the coculture, while *S. sanguinis* viability in the presence of *S. mutans* was below the limit of detection (see Fig. S1 in the supplemental material). Similarly, when equal volumes of *S. mutans* and *S. gordonii* were incubated for 24 h, *S. mutans* showed a 39-fold increase in growth ($P < 0.001$), while *S. gordonii* exhibited only a 3-fold increase (Fig. S1). Interestingly, when grown in medium supplemented with the supernatant from *S. mutans* cultures, the viability of *S. sanguinis* and *S. gordonii* was equally impaired. These results indicate that the inhibition of viability or growth is mediated by secreted factors present in the medium (Fig. S1) and support the idea that the opportunistic growth of *S. mutans* is not due to a growth kinetic advantage, since the commensals grew when cultured alone in the same time frame. Overall, these data suggest that *S. mutans* can outcompete both *S. sanguinis* and *S. gordonii*, consistent with previous research on the competition between *S. sanguinis* and *S. mutans* in the oral cavity (25–28).

To understand the effect of e-cig aerosols on *S. sanguinis* and *S. gordonii* growth pattern, we grew the bacteria in TSB medium pre-exposed to e-cig aerosol overnight and recorded bacterial turbidity every 5 min as the optical density at 600 nm ($OD_{600}$). We observed that upon exposure to e-cig aerosol-pretreated medium, the growth of *S. sanguinis* was negatively affected, reaching a lower plateau than the untreated control (Fig. 1A). The most significant reduction of biomass accumulation was observed under conditions containing the combination of nicotine and menthol flavoring, as seen by the lowest plateau curve. Nicotine-free aerosol overall affected growth compared to the untreated control but was less harmful than the aforementioned condition. The bacterial growth rates were plotted as the linear regression of the exponential phase (Fig. 1B; Fig. S2). While the growth rates remained the same when these commensals were exposed to aerosol, the overall biomass accumulation was lower for all the e-cig aerosol treatments than the control. Overall, e-cig aerosols with and without nicotine and with and without menthol flavor all significantly hindered the bacterial biomass accumulation of *S. sanguinis* at 4, 6, and 8 h, compared to the control (Table 1).

Similarly, exposure of *S. gordonii* to e-cig aerosol-pretreated medium resulted in reduced biomass accumulation under all conditions. Growth in nicotine-containing aerosol and, more so, the combination of nicotine and menthol flavoring reached a lower plateau than the other conditions (Fig. 1C). *S. gordonii* shows a biphasic growth curve characterized by two exponential growth phases that are separated by a lag phase. We plotted the linear regression of the first exponential phase, which did not show significant changes between the treatment groups and the untreated control (Fig. 1D; Fig. S2). However, all four treatments hindered the growth of *S. gordonii*, as observed by the disorganized growth patterns and the lower growth plateau under these conditions. All the aerosol treatments significantly decreased the biomass accumulation of *S. gordonii*, observed by the lower $OD_{600}$ at 8, 12, 14, and 16 h than that of the control (Table 1).

When e-liquid was added to the bacterial growth medium without heating and aerosolization, only a minor reduction in bacterial growth for both *S. sanguinis* and *S. gordonii* was observed (Fig. S3). This indicates that the heat generated by the atomizer heating coil results in the emission and inhalation of harmful and toxic vape by-products, as seen in Fig. 1. These results support previous reports that nonaerosolized e-liquid does not affect the growth of *Streptococcus* spp. tested, including *S. gordonii* (20,

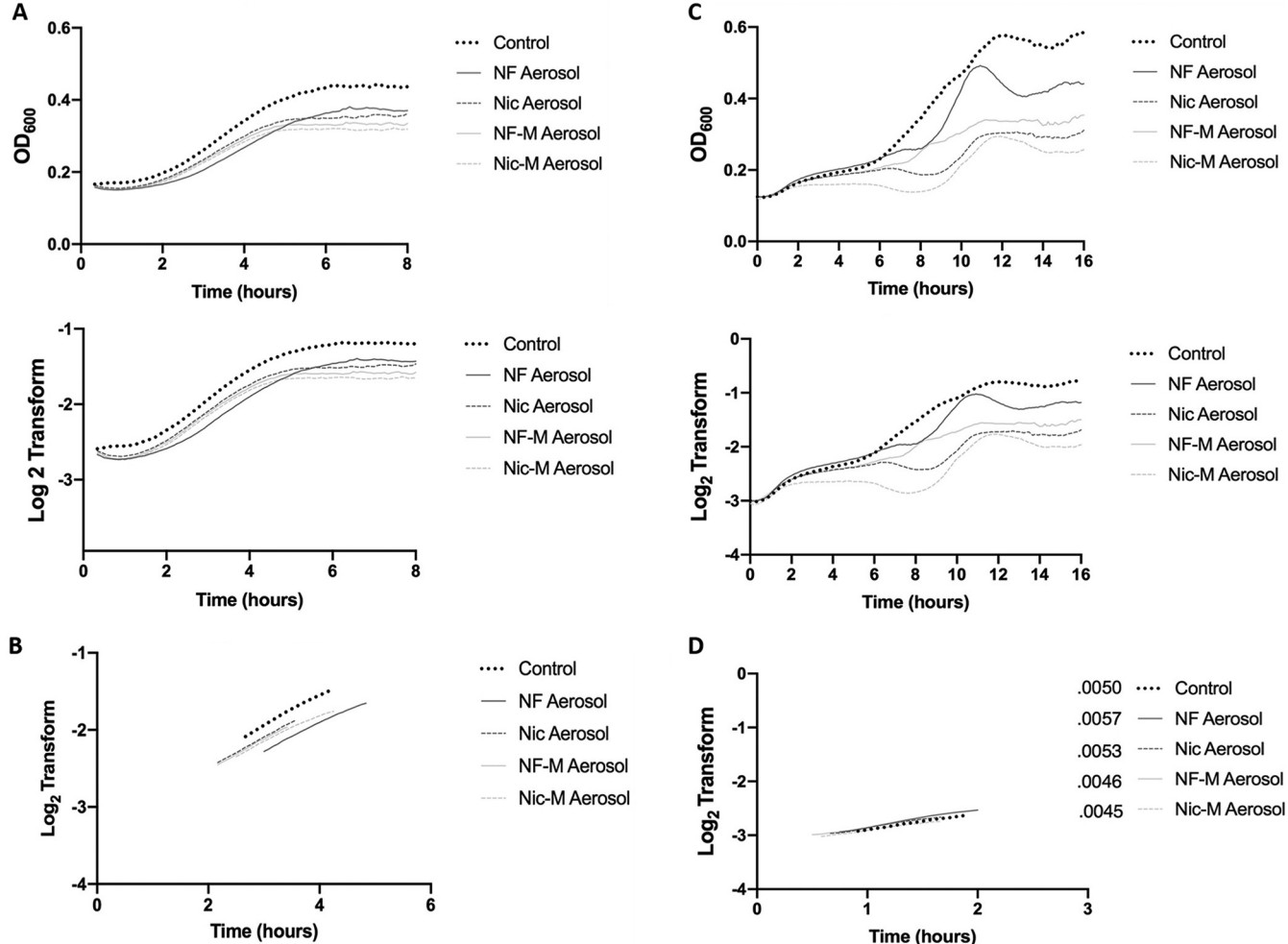

**FIG 1** E-cigarette aerosols reduce the growth of *S. sanguinis* and *S. gordonii*. (A) Overnight cultures of *S. sanguinis* were diluted 1:10 into e-cig aerosol-pretreated medium. $OD_{600}$ was measured every 5 min over the course of 8 h. The average was calculated and plotted for $OD_{600}$ (top) and $\log_2$ transform (bottom). (B) Slopes for linear regression of *S. sanguinis* are presented based on the derivative of the $\log_2$ transform. (C) Overnight cultures of *S. gordonii* were diluted 1:10 into e-cig aerosol-pretreated medium. Optical density was measured every 5 min over the course of 16 h. The average was calculated and plotted $OD_{600}$ (top) and $\log_2$ transform (bottom). (D) Slopes for linear regression of *S. gordonii* are presented based on the derivative of the $\log_2$ transform. Data are representative of one biological ($n = 3$) replicate repeated at least twice. $OD_{600}$ of treatment groups was compared to control $OD_{600}$, using one-way ANOVA (Dunnett's correction). All conditions were compared to the control; for a complete breakdown of significance, refer to Table 1.

22). Oxidation and decomposition upon heating of the universal carrier components, glycerol and propylene glycol (4), can result in the formation of formaldehyde and acetaldehyde (29). It has been reported that these chemicals are bactericidal and have a negative effect on the oral microbiota (30).

Next, we explored the effect e-cig aerosols have on the known oral pathogen *S. mutans*. The experiment was performed the same way as described for *S. sanguinis* and *S. gordonii*. *S. mutans* growth in the different conditions was recorded. The growth curve for *S. mutans* shows no changes or $OD_{600}$ reduction compared to the control (Fig. 2A), suggesting resistance to e-cig aerosols compared to the streptococcal commensals. Furthermore, the linear regression of the exponential phase is almost identical, as seen from the overlapping slopes (Fig. 2B; Fig. S2). No changes were observed when *S. mutans* was exposed to e-liquid alone (Fig. S3). Overall, we can conclude that *S. mutans* growth remains unaffected when exposed to e-cig aerosols regardless of nicotine content or flavoring.

Additionally, we compared the linear regression of the three species when they were exposed to the different e-cig aerosols. We observed significantly higher growth rates for

**TABLE 1** Biomass analysis for the different vape conditions[a]

| Organism | Condition | P at time (h) | | | | | | |
|---|---|---|---|---|---|---|---|---|
| | | 4 | 6 | 8 | 10 | 12 | 14 | 16 |
| S. sanguinis | NF | <0.0001 | 0.0011 | 0.0002 | | | | |
| | Nic | 0.0002 | 0.0005 | 0.0003 | | | | |
| | NF-M | <0.0001 | 0.0002 | <0.0001 | | | | |
| | Nic-M | <0.0001 | <0.0001 | <0.0001 | | | | |
| S. gordonii | NF | NS | NS | 0.0342 | NS | 0.0056 | 0.0228 | 0.0107 |
| | Nic | NS | 0.0169 | 0.0029 | 0.0036 | <0.0001 | 0.0006 | 0.0003 |
| | NF-M | NS | 0.0404 | 0.0277 | 0.0123 | 0.0002 | 0.0010 | 0.0007 |
| | Nic-M | <0.0001 | <0.0001 | 0.0002 | 0.0008 | <0.0001 | 0.0001 | <0.0001 |
| S. mutans | NF | NS | NS | NS | | | | |
| | Nic | NS | NS | NS | | | | |
| | NF-M | NS | NS | NS | | | | |
| | Nic-M | NS | NS | NS | | | | |

[a]$OD_{600}$ values at each time point were compared to the control $OD_{600}$ using a one-way ANOVA (Dunnett's correction). Significance identifies $OD_{600}$ values that are lower than the control, which represents a lower biomass. NS, not significant.

S. mutans than S. sanguinis and S. gordonii under all conditions (Fig. 3). These results highlight how the impact of e-cig aerosols greatly differs between bacterial strains and could ultimately lead to an imbalance by suppressing the growth of the commensals, thereby providing a growth advantage to S. mutans. We speculate that these opposing effects could lead to changes in oral bacterial homeostasis of e-cig users.

**E-cigarette aerosol exposure results in changes in hydrophobicity and promotes biofilm formation of S. mutans compared to S. sanguinis and S. gordonii.** Bacterial biofilms are found on most surfaces in the oral cavity and are often formed as a response to an environmental challenge. As such, they ensure that bacteria can survive in a protected environment, resisting eradication (31, 32). Therefore, changes in biofilm formation upon e-cig aerosols could affect the composition of the oral streptococcal community. We measured biofilm formation of S. sanguinis (Fig. 4A), S. gordonii (Fig. 4B), and S. mutans (Fig. 4C) when grown in e-cig aerosol-pretreated medium without physical disruptions for 24 h. The biofilm was stained with safranin and quantified by measuring $OD_{490}$. Our results show that S. sanguinis and S. gordonii biofilm formation was unaffected when the bacteria were exposed to e-cig aerosol treatments regardless of the presence of nicotine or flavorings (Fig. 4A and B). As reported in the literature, S. sanguinis and S. gordonii utilize sucrose as a carbon source to synthesize glucan, which is an important component of their biofilms (33). We therefore performed biofilm assays for these commensals in medium supplemented with 1% sucrose along with e-cig exposure to determine if the more physiological conditions would enhance the resistance to e-cig aerosols. Our results showed that while the overall capacity for biofilm formation was enhanced in the presence of sucrose, bacterial biofilm formation in response to e-cig vape was not affected compared to the sucrose-supplemented control (Fig. S4).

We observed a significant increase in biofilm formation when S. mutans was exposed to nicotine-free aerosol, nicotine aerosol, nicotine-free menthol aerosol, and nicotine menthol aerosol, all compared to untreated control (Fig. 4C). E-liquid exposure for 24 h showed only minimal effects on S. mutans biofilm formation (Fig. S5), again stressing the importance of heating the e-liquid and aerosolizing the components. We surmise that the observed increase in biofilm formation upon e-cig exposure could potentially confer an advantage to S. mutans by allowing protection and subsequently further colonization of the oral cavity.

Several bacterial properties can be an indication for the ability to adhere to inert surfaces, one of them being hydrophobicity (34). Additionally, bacterial hydrophobicity has been correlated with changes in biofilm formation (35). As bacteria can switch their membrane phenotype between hydrophilic and hydrophobic due to environmental

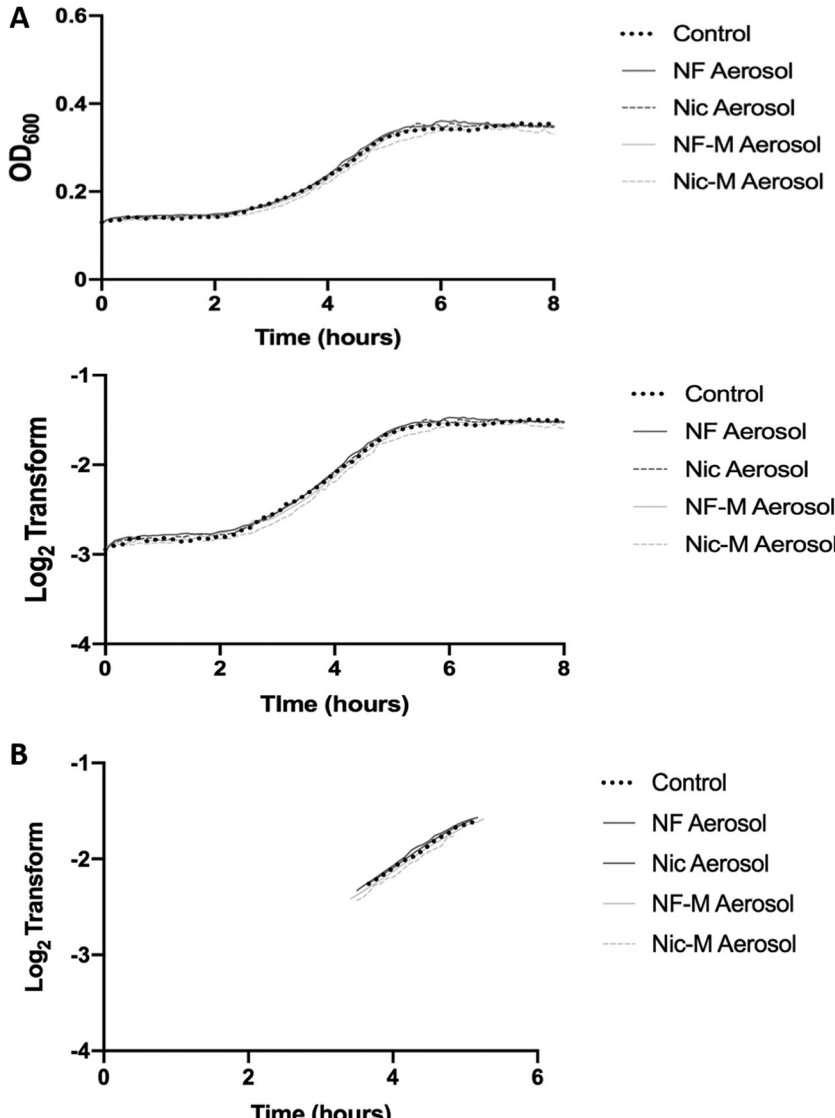

**FIG 2** E-cigarette aerosols did not affect the growth patterns of *S. mutans*. (A) Overnight cultures of *S. mutans* were diluted 1:10 into e-cig aerosol-pretreated medium. Optical density was measured every 5 min over the course of 8 h. The average was calculated and plotted for $OD_{600}$ (top) and $\log_2$ transform (bottom). (B) Slopes for linear regression of *S. mutans* are presented based on the derivative of the $\log_2$ transform. Data are representative of one biological replicate ($n = 3$) repeated at least twice.

changes (36), we wanted to explore the effect of e-cig aerosol exposure on the hydrophobicity of these streptococcal strains. *S. sanguinis* had a significant decrease in hydrophobicity when exposed to nicotine and nicotine-free menthol (Fig. 5A). *S. gordonii* (Fig. 5B) displayed a lower hydrophobicity overall. Interestingly, the hydrophobicity of *S. mutans* was overall higher than that of the commensals and remained around 65% when the organisms were exposed to e-cig aerosol (Fig. 5C) with a significant increase in nicotine-free conditions. To visualize how the hydrophobicity could affect growth patterns of the streptococcal strains in the presence of a biotic surface, we cultured immortalized oral epithelial OKF6 cells with each of the streptococci at a multiplicity of infection (MOI) of 10 for 1 h. At the end of the coculture, we used crystal violet to visualize the attached bacteria by bright-field microscopy (Fig. 5D). All three strains were able to adhere to OKF6 cells, yet *S. mutans* had the highest propensity to coaggregate, which relates back to its highest hydrophobicity.

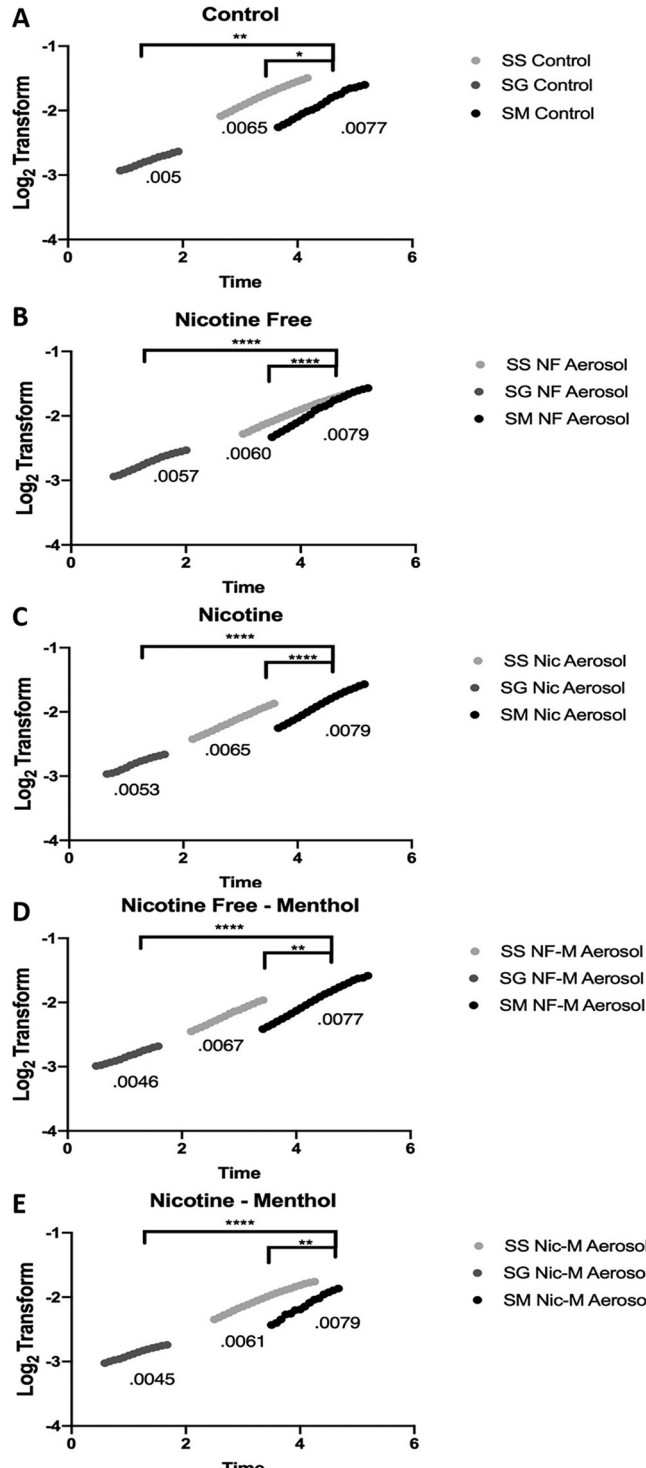

**FIG 3** *S. mutans* has a significantly higher growth rate than the commensal streptococcal species. Overnight cultures of *S. sanguinis* (SS), *S. gordonii* (SG), and *S. mutans* (SM) were diluted 1:10 in TSB (A) or e-cig aerosol that was nicotine free (B), nicotine containing (C), nicotine free with menthol (D), or nicotine containing with menthol (E). Optical density was measured every 5 min, and growth curves were analyzed. Slopes for linear regression are presented based on the derivative of the averaged $\log_2$ transform. Data are representative of one biological (*n* = 3) replicate repeated at least twice. For each condition, the commensal's growth rate was compared to that of *S. mutans* using one-way ANOVA (Dunnett's correction). *, $P < 0.05$; **, $P < 0.01$; ****, $P < 0.0001$.

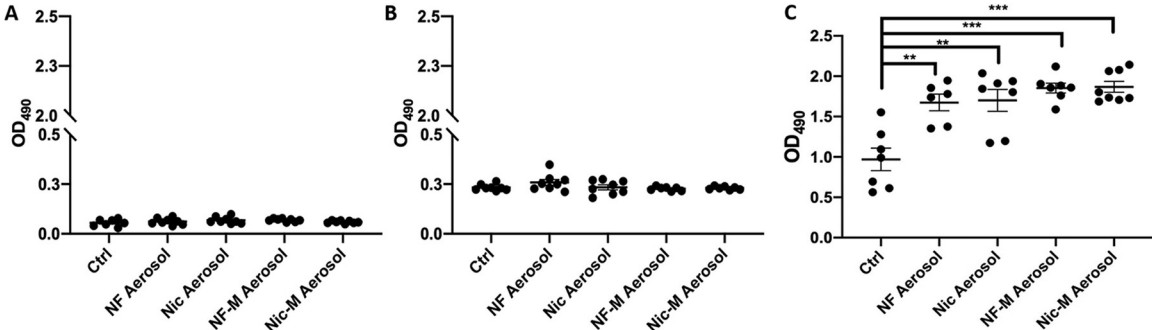

**FIG 4** E-cigarette aerosols promote biofilm the formation of *S. mutans* but not *S. sanguinis* and *S. gordonii*. Overnight cultures of *S. sanguinis* (A), *S. gordonii* (B), and *S. mutans* (C) were diluted 1:10 in e-cig-pretreated medium and grown for 24 h. Biofilms were stained with safranin, and $OD_{490}$ was measured. The data are means and SEM ($n$ = 7) for one biological replicate of 3. Groups were compared to the control using one-way ANOVA (Dunnett's correction). **, $P < 0.01$; ***, $P < 0.001$.

***S. mutans* exhibits higher attachment to oral epithelial cells even when challenged with E-cig aerosols compared to *S. sanguinis* and *S. gordonii*.** Attachment to epithelial cells is an essential aspect of bacterial colonization in the oral cavity (36). We therefore aimed to identify the differences in attachment capabilities of the three model strains after exposure to e-cig aerosol. To quantify bacterial attachment to oral epithelial cells, we cultured OKF6 cells with each of the streptococci at an MOI of 1 for 3 h and enumerated CFU for control and e-cig aerosol-exposed cocultures. Although we did not observe significant differences between the treated and control groups for each species, we consistently observed that *S. mutans* exhibits a higher attachment capacity than the two commensal strains (Fig. 6). Bacterial attachment to the mucosal membrane can lead to inflammation and other systemic diseases (18, 37). In many instances, microbial adhesion, aggregation, and biofilm formation cause serious damage and disease (36). We have demonstrated that *S. mutans* exhibits all these characteristics.

## DISCUSSION

E-cigs were introduced to the United States and Europe 15 years ago, but their effects on oral health have not been well established yet (4). Tobacco smoking results in significant shifts in the oral microbiome, as observed in patients with oral cancer (38). Similar to conventional cigarette smoke, e-cigs are known to cause dysbiosis by affecting the growth of oral commensals, allowing opportunistic pathogens to grow (1). However, the role e-cigs play in the behavior and survival of specific bacterial species in the oral cavity remains unclear. Our study explored the effect of e-cigs on three *Streptococcus* spp. commonly found in the oral cavity. We demonstrated how e-cig aerosol exposure induces differential responses depending on the strain of oral bacterium, potentially contributing to changes in oral bacterial composition.

To better understand how e-cig aerosols affect the oral resident bacteria, we exposed three *Streptococcus* spp. to e-cig aerosols with and without nicotine or flavoring and measured growth patterns, biofilm formation, hydrophobicity, and attachment to oral epithelial cells. Our results show that e-cig aerosols disrupt the growth patterns of *S. sanguinis* and *S. gordonii*, without affecting the growth of *S. mutans*. E-cig devices utilize e-liquid, which generally contains propylene glycol and vegetable oil to generate vapor and acts as a carrier for additives such as nicotine and flavors (39). Propylene glycol itself has been shown to have a bactericidal effect and can therefore modulate the oral microbiota (30). These reports support our finding of overall lower biomass accumulation in even the nicotine-free and flavorless condition. Interestingly, few pathogens have been shown to be resistant to propylene glycol, one of them being *Staphylococcus aureus* (30); we report here that *S. mutans* is resistant as well.

Nicotine has been shown to support *S. mutans* oral colonization by increasing biofilm formation and viability of *S. mutans* (40). More so, nicotine concentrations at 1 to 4 mg/mL upregulate the expression of virulence receptor proteins and extracellular

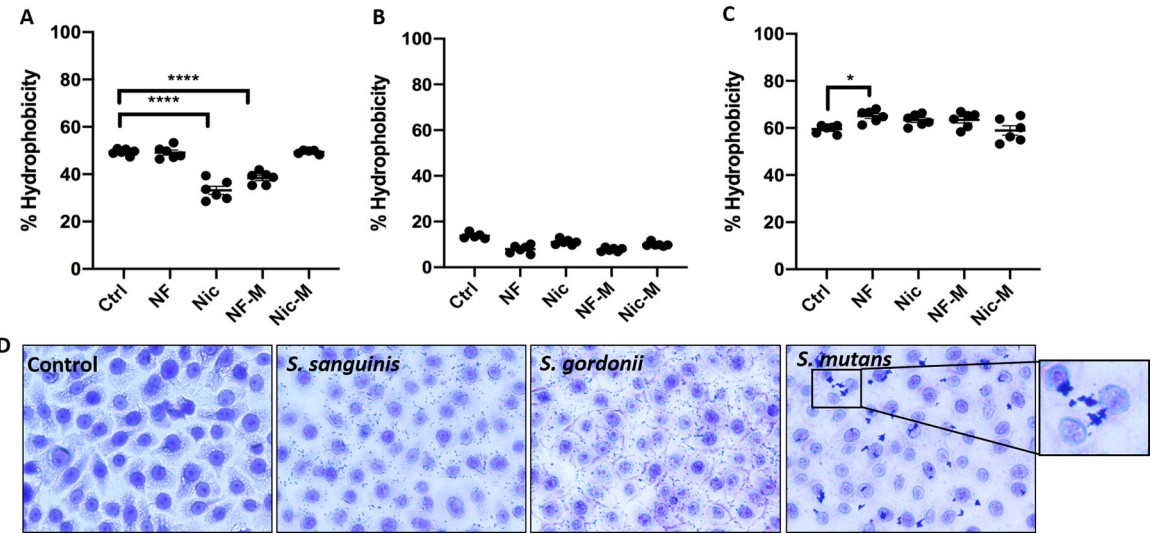

**FIG 5** *S. mutans* shows a higher hydrophobicity and coaggregation capacity than *S. sanguinis* and *S. gordonii*. Overnight cultures of *S. sanguinis* (A), *S. gordonii* (B), and *S. mutans* (C) were diluted 1:10 in e-cig-pretreated medium and grown for 24 h. Chloroform was added to each condition, and after incubation, the aqueous layer was collected and $OD_{600}$ was measured. The data are means and SEM ($n = 3$) for three biological replicates. Groups were compared to controls using one-way ANOVA (Dunnett's correction). *, $P < 0.05$; ****, $P < 0.0001$. (D) Bright-field microscopy of crystal violet-stained 1-h cocultures (MOI, 10) of oral epithelial cells (OKF6) with *S. sanguinis*, *S. gordonii*, and *S. mutans*. Magnification, ×40.

polysaccharides, ultimately stimulating glycolytic pathway intermediates (41, 42). Overall, it has been reported that e-cig aerosols enhance adherence of *S. mutans* to the teeth and the ability to form biofilms (43). Mechanistically, e-cig aerosols regulate the expression of biofilm-associated genes in *S. mutans*. Specifically, *comC, -D*, and *-E*, which are related to quorum sensing genes, are central to cell viability and biofilm formation in response to environmental conditions (43). Our data further support the idea that *S. mutans* may be at an advantage when environmental stressors such as e-cig aerosol exposure arise in the oral cavity.

Bacteria contain cell surface hydrophobicity (CSH) features, including hydrophobic amino acid residues, outer membrane proteins and lipids, and lipoteichoic acid (36). Interestingly, bacteria can switch their membrane phenotypes between hydrophilic and hydrophobic due to environmental changes (36). The importance of hydrophobic measurements is supported by the findings that patients with a high risk for dental caries have a higher prevalence of hydrophobic bacteria than patients with low dental caries risk (44). We assessed hydrophobicity, which has also been correlated with adhesion (36) but is only one factor associated with adhesion and attachment to surfaces in the context of e-cig aerosol exposure. We observed that e-cig aerosol exposure reduced the hydrophobicity of *S. sanguinis*, while the hydrophobicity of *S. gordonii* remained low. On the other hand, e-cig aerosols did not affect the higher hydrophobicity of *S. mutans*. More interestingly, we observed a tendency in *S. mutans* to coaggregate. Overall, the higher hydrophobicity and coaggregation capacities observed for *S. mutans* could be correlated with tissue invasion, inflammation, and disease (36).

The same oral epithelial cells (OKF6) we used in this study were subjected to biofilm and planktonic *S. gordonii* (45) challenge by Ebersole et al. (46). Their findings elucidated major gene differences of epithelial cells in the presence of biofilm versus planktonic *S. gordonii*. The highest upregulation of immune-related genes was found with the biofilm group and included important cytokines involved in the innate and adaptive immune response, including interleukin 8 (IL-8; CXCL8). *S. mutans*, in a biofilm challenge experiment using gingival cells, stimulated the expression of IL-8 without other bacteria present but did not induce antimicrobial peptides such as DEFB4B. In contrast, coculture with *S. mitis*, which is involved in early plaque formation, resulted in the expression of antimicrobials, including DEFB4B, but did not influence inflammatory

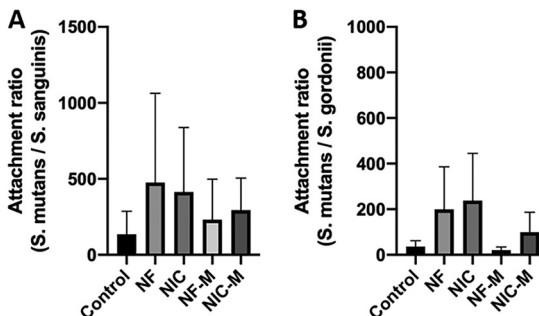

**FIG 6** *S. mutans* has a higher capacity to adhere to oral epithelial cells than *S. sanguinis* and *S. gordonii*. OKF6 cells were cocultured with *S. sanguinis*, *S. gordonii*, and *S. mutans* for 3 h at an MOI of 1. OFK6 cells with possible internalized and attached bacteria were permeabilized and serial dilutions plated for CFU/mL counts. For each condition, the attachment of *S. mutans* was compared to that of *S. sanguinis* (A) or *S. gordonii* (B) by determining the ratio of *S. mutans* to the commensal. (SEM; *n* = 3). Groups were compared to control using one-way ANOVA (Dunnett's correction).

gene expression (46). This study highlights another mechanism of *S. mutans* oral colonization: while bacteria involved in early plaque formation can induce the synthesis of antimicrobial peptides, bacteria like *S. mutans* that are secondary colonizers are not recognized as pathogenic by epithelial cells and subsequently evade the immune response, which may allow opportunistic colonization (46).

We evaluated the interaction of the selected streptococcal strains with OKF6 cells measuring attachment only and observed that *S. mutans* overall had a higher capacity to adhere to OKF6 cells than the commensals. While the results from the attachment assay correlated with biofilm formation and hydrophobicity, the mechanisms for differences in adhesions need to be further determined. Specific surface proteins called adhesins can be involved in the binding of bacteria to human cells (47). For example, *S. gordonii* expresses the surface proteins of the antigen I/II adhesin family, SspA and SspB. These adhesins bind directly to $\beta$1 integrin and mediate internalization (48). It has also been reported that *Streptococcus gordonii*, *Streptococcus sanguinis*, and *Streptococcus oralis* (49) have surface appendages that play a role in the adhesion of streptococci to host cells but also in interbacterial aggregation (50, 51). These findings raise the questions of whether surface marker expression is distinct in the three model bacteria we selected and whether e-cig vape could modify the cell membrane composition of the bacteria. Our investigation stops short of identifying mechanisms of interbacterial binding and adhesion to epithelial cells, but it highlights the ideas that ligand-binding specificities of surface proteins may account for species-specific adherence within the human oral cavity and that the effects of e-cig aerosols can differ between strains. This could be the reason behind differences seen between the three species and their attachment.

Given that 80% of the oral cavity microbiota is composed of *Streptococcus* spp., our study is proof of concept for how e-cig exposure can alter diverse behaviors in the three model organisms we analyzed, affecting commensals differently from opportunistic pathogens. By extension, we propose that e-cig vaping can modify the oral microbiome differentially. In homeostasis, *S. sanguinis* and *S. gordonii* are early colonizers (18). As such, they protect the oral cavity from colonization by pathogens like *S. mutans* (18). In addition, *S. mutans* is a known opportunistic pathogen, and its role in dental caries and periodontitis has been well established (23, 24). When environmental changes such as e-cig vapor exposure disrupt the oral microbial homeostasis, the decrease in commensals can allow opportunistic pathogens to colonize and cause disease (52). The repetitive exposure to anthropogenic stressors, like e-cigs, can exert an evolutionary pressure that can lead to the dysregulation of the oral bacterial homeostasis. This homeostasis is important because prevalent oral diseases, including dental caries, gingivitis, and periodontitis, are driven by bacteria and their capacity to form dental plaque (51). A major limitation of this study is the restriction to three organisms,

when the oral cavity is colonized with complex communities. To expand on that and to provide a more physiological model of *Streptococci* spp. in the oral cavity, more *in vivo* and *in vitro* multispecies studies will be required in the future. In addition, more research needs to be conducted regarding interspecies interactions and the long-term consequences e-cig aerosols could have on host-bacterial interactions with further analysis of host responses.

Based on our data, we can merely speculate that e-cig aerosol exposure can shift the abundance of bacteria present in the oral cavity, as seen with three key species, and result in significant consequences for oral health.

## MATERIALS AND METHODS

**Cell lines.** Human oral epithelial cells (OKF6) (53) were cultured in keratinocyte serum-free medium (10724-011; Life Technologies Co., Grand Island, NY, USA), supplemented with epithelial growth factor (1 ng/mL), bovine pituitary extract (0.05 mg/mL), and 1% penicillin-streptomycin (15140-122; Gibco Life Technologies Co., Carlsbad, CA, USA). Cells were incubated at 37°C with 5% $CO_2$.

**Bacterial cultures.** *Streptococcus mutans* (ATCC 25175), *Streptococcus sanguinis* (ATCC 10556), and *Streptococcus gordonii* (ATCC 51656) were all grown in tryptic soy broth (TSB; 211825; Becton Dickinson, Sparks, MD, USA) at 37°C and 5% $CO_2$. Bacteria were grown in tryptic soy agar (TSA) plates for standard serial dilution plating. Bacteria were subcultured in TSB from an independent colony, snap frozen in liquid nitrogen for 30 min during the exponential growth phase, and stored at −80°C. To determine the number of CFU per milliliter, bacteria were serially diluted in phosphate-buffered saline (PBS) and spot plated (20 $\mu$L) in TSA.

**Competition assay.** Overnight cultures were diluted 1:10 in TSB. The controls contained only one species, while the competitive growth tubes contained 1 part *S. mutans* and 1 part commensal (either *S. sanguinis* or *S. gordonii*). At 0 h and 24 h, each tube was serially diluted and plated on blood agar medium via spot inoculations, and the number of CFU/mL was calculated for each time point. Hemolysis patterns were analyzed to differentiate between the two *Streptococcus* spp. *S. mutans* was identified on the basis of beta hemolysis, while *S. sanguinis* and *S. gordonii* were identified via alpha hemolysis. Competitive growth data were shown as fold change by calculating the ratio between the 24-h and the 0-h time points. Regarding the supernatant exposure assay, *S. sanguinis* and *S. gordonii* were grown in TBS supplemented with 1 mL PBS (control group) or 1 mL of *S. mutans* supernatant. After 24 h, numbers of CFU were determined. Fold changes was determined based on the CFU/mL data.

**E-cigarette aerosol exposure.** Commercially available e-liquid was purchased from Vapor Vapes, all prepared with a base of 50% propylene glycol and 50% vegetable glycer (50/50 PG/VG). In this study, we had four experimental groups: the control PG/VG without nicotine or flavoring (NF), the nicotine control with PG/VG and nicotine (3 mg/mL) (Nic), the control flavored PG/VG (menthol) without nicotine (NF-M), and the PG/VG with menthol and nicotine (3 mg/mL) (Nic-M). These four e-liquid combinations were heated by a fourth-generation e-cig (G-Priv Baby kit) and vaporized (one 10-s puff, 5-min exposure) into TSB medium (10 mL in a 10-cm dish) before every experiment.

**Growth curve.** TSB was pretreated with e-cigarette aerosols (one 10-s puff, 5-min exposure) before every experiment. Overnight cultures were diluted 1:10 in pretreated medium and loaded in a 96-well plate (100 $\mu$L per well) incubated for 24 h at 37°C in a Biotek Synergy plate reader. During this incubation, optical density (600 nm) readings were done every 5 min. Growth curve data were graphed and analyzed on GraphPad (Prism 8).

**Biofilm formation.** TSB was pretreated with e-cigarette aerosols (10-s puff, 5-min exposure), and overnight cultures were diluted 1:10 in pretreated medium and incubated for 24 h in a 96-well plate (100 $\mu$L per well). After 24 h of incubation, plates were washed with deionized water and air dried at 37°C for 20 min. To stain the biofilm, safranin (65092B-95; Millipore Sigma, Darmstadt, Germany; 125 $\mu$L containing 6.04g/L safranin, 19% ethanol, and 1% methanol, diluted in water) was added to the wells and incubated at room temperature for 20 min. Safranin was discarded, and the wells were washed with deionized water and air dried at 37°C. The biofilm was dissociated by resuspending in ethanol-acetone mix (80:20). The absorbance was read at $OD_{490}$. Results were normalized to blanks, graphed, and analyzed on GraphPad Prism (Prism 8).

**Hydrophobicity assay.** To measure bacterial surface hydrophobicity, we carried out a hydrophobicity assay (54), as previously described by Serebryakova et al. (54) with slight modifications. Bacterial cultures at stationary phase (2 mL) were mixed with chloroform (500 $\mu$L), vortexed for 2 min, and incubated at 37°C for 15 min to allow separation between the aqueous and hydrophobic phases. During this separation, the optical density of the aqueous (hydrophilic) phase decreases, as the hydrophobic bacteria moves into the chloroform (hydrophobic) phase. The aqueous layer ($A_a$) was collected, and $OD_{600}$ was measured. The $OD_{600}$ of the overnight culture ($A_t$) was measured to account for the total bacteria before the phase separation. The percent hydrophobicity was calculated as follows: $[(A_t - A_a)/A_t] \times 100$.

**Bacterial attachment.** To visualize bacterial attachment to oral human cells, cells were seeded at a confluence of 300,000 cells/well in a 4-chamber slide (154526; Nalge Nunc International, Rochester, NY, USA). Bacterial cells were added to the slide at an MOI of 10. After the 3-h coculture, the slide was rinsed extensively with PBS to remove any unattached bacterial cells and then fixed with methanol and acetone (1:1) for 10 min. Cells were then stained with crystal violet and visualized with a bright-field microscope. For quantification of attachment and internalization, oral human cells were seeded at a

confluence of $2.5 \times 10^5$ cells/mL in 24-well plates and cocultured with e-cig-pretreated streptococcal strains at an MOI of 1 for 3 h. At the end of the incubation time, OKF6 cells were washed with PBS, and the remaining cells with attached and potentially internalized bacteria were permeabilized with 1% Triton X-100 for 10 min followed by a 1:10 dilution with sterile PBS. Next, the lysates were serially diluted and plated to enumerate CFU/mL. The final attachment counts were normalized to the initial number of input bacteria (both in CFU/mL) per well. For each experimental condition, counts of attached *S. mutans* bacteria were compared to those of both commensals by calculating the ratios of *S. mutans* to *S. sanguinis* and of *S. mutans* to *S. gordonii*. This ratio indicates the fold change between the attachment capacities of the commensal and pathogenic strains.

**Analysis and statistics.** Experiments were performed in biological replicates, each consisting of at least three technical replicates. The data are presented as means and standard errors of the means (SEM; $n = 3$). When comparing only two conditions, we used Student's $t$ test (Welch's correction) to analyze statistically significant differences. When comparing multiple conditions to the untreated control, we used one-way analysis of variance (ANOVA) (Dunnett's correction) to analyze statistical significance differences.

## SUPPLEMENTAL MATERIAL

Supplemental material is available online only.

**SUPPLEMENTAL FILE 1**, PDF file, 0.8 MB.

## ACKNOWLEDGMENTS

This work was supported by a COM grant to C.D.A. and by F31DE029102 to A.C.-V.

We have no conflicts of interest to declare.

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
