## [Reviewer comments · Microbiology Spectrum]

Microbiology Spectrum

E-cigarette aerosol exposure favors the growth and colonization of oral *S. mutans* compared to commensal Streptococci

Alma Catala-Valentın, Josh Bernard, Matthew Caldwell, Jessica Maxson, Sean Moore, and Claudia Andl

Corresponding Author(s): Claudia Andl, University of Central Florida

Review Timeline:

Submission Date:	December 2, 2021
Editorial Decision:	December 21, 2021
Revision Received:	February 21, 2022
Accepted:	March 2, 2022

Editor: Justin Kaspar

Reviewer(s): The reviewers have opted to remain anonymous.

Transaction Report:

DOI: <https://doi.org/10.1128/spectrum.02421-21>

December 21, 2021

Dr. Claudia D. Andl
University of Central Florida
Burnett School of Biomedical Sciences
4111 Libra Dr
Orlando, FL 32816

Re: Spectrum02421-21 (E-cigarette aerosol exposure promotes dysbiosis by favoring the growth and colonization of oral S. mutans)

Dear Dr. Claudia D. Andl:

Two expert reviewers have provided opinions on your manuscript. A general conclusion from both reviewers are that the broad conclusions of the work as presented are not supported by the current results shown. Please refer to the comments of reviewer 1 - I agree with the view point of this reviewer that the limited observations fall short of the conclusions as currently described in the abstract, importance and discussion sections. However, many of the additional comments can be easily addressed.

I would happily consider a revision of this manuscript that tones down the conclusions and solely focuses on the description of growth and biofilm formation of single oral streptococci species in the presence of e-cig vapor (current data provided) that does not include comments regarding the entire oral microbiome and dysregulation of homeostasis. This could be led into for future directions regarding this project, but not as a result of this current study. For conclusions on the shifting of the supragingival community in the presence of e-cig vapor to remain, dual- and/or multi-species experiments would be necessary (additional experiments would be required - again refer to reviewer 1). A revision that maintains the current conclusions with the current results and only addresses the few other review comments would not be viewed favorably.

Link Not Available

Sincerely,

Justin Kaspar

Journals Department
Reviewer comments:

Reviewer #1 (Comments for the Author):

The article by Catala-Valentin and collaborators aim to investigate the effect of e-cig vapor on the oral microbiome and its

potential implication on oral health. The topic is of extreme importance, but I think the work presented here does very little to address the authors' goal, and I would certainly disagree that the authors "demonstrated how e-cig aerosol could shift the model community to be dominated by *S. mutans*, contributing to a dysregulation in the bacterial homeostasis." I think the authors presented an interesting observation but fell short of explaining how e-cigs contribute to their observation.

Specific points -

Introduction - The authors dedicate an entire paragraph to e-cig and periodontal disease. Still, the authors focus on *S. mutans*, which is associated with tooth decay and not with periodontal disease. This seems puzzling to me. Additionally, there does not seem to be any evidence of increased tooth decay due to the use of e-cigs. Thus, based on the introduction, I do not understand why the authors focus on *S. mutans*.

Additionally, It is unclear how focusing on three strep species grown as single species will help you better understand the oral microbiome. Maybe you should consider doing multi-species biofilms to see how e-cigs affect these streps when grown under slightly more complex conditions than just pure culture.

Methods -

Bacterial culture- Minor typo: it should be bacteria were sub-cultured in TSB, not "bacteria was..."

Growth curve - I am particularly interested in the exposure of TSB to e-cig vapors. The authors mention that TSB was pretreated with e-cig aerosol (10-sec puff, 5-minute exposure). Was this done only once, or were this repeated multiple times? As written, it is unclear to me.

Biofilm formation - *sanguinis* and *gordonii* are lousy biofilm formers in the absence of a saliva-coated surface or sugar, like sucrose. To get a little closer to oral conditions, I would encourage the authors to coat their 96-well plates with saliva.

Results -

The authors are trying to make the case that a decrease in *S. s* and *S. g* could favor the growth of *S. m* in the oral cavity. The authors should consider co-culturing the different steps and exposing the multi-species biofilm to e-cig vapor to determine any effect of the growth of the different species.

Page 12 - minor typo. I think the authors may want to say conducive instead of conductive.

Page 13 - The authors overstate the link between hydrophobic and host surface attachment, neglecting the fact that interactions with host surfaces can also be mediated by bacterial surface proteins. I encourage the authors to tone their conclusion down here because while hydrophobicity may contribute to surface attachment, it is not the only factor.

Fig 5B - It is not clear to me what I learned from the experiment shown in fig 5B. There's no quantification (for example, CFU of OKF6 adhered bacteria), nor did the authors investigate the effect of e-cig vapor on bacterial adhesion to OKF6 cells. So what is the point of this experiment?

Reviewer #2 (Comments for the Author):

In this study, Catala-Valentin et al. studied the impact of E-cigarette aerosol on the growth and biofilm formation of three oral streptococci. They showed that E-cigarette aerosol inhibited the growth of two commensal *Streptococcus* species, i.e., *S. sanguinis* and *S. gordonii*, but did not affect the growth of an oral pathogen, *S. mutans*. Most of the results shown in this study have been reported by previous publications. In the papers PMID: 31835369 and PMID: 33329035, the authors showed that the growth of oral commensal streptococci were inhibited by E-cigarette aerosol. In PMID: 32683796, E-cigarettes increased the growth and biofilm formation of *S. mutans* on teeth surfaces. More similar publications can be found. Hence, the authors need to show more innovative results.

Staff Comments:

Preparing Revision Guidelines

To submit your modified manuscript, log onto the eJP submission site at <https://spectrum.msubmit.net/cgi-bin/main.plex>. Go to Author Tasks and click the appropriate manuscript title to begin the revision process. The information that you entered when you first submitted the paper will be displayed. Please update the information as necessary. Here are a few examples of required

updates that authors must address:

Please return the manuscript within 60 days; if you cannot complete the modification within this time period, please contact me. If you do not wish to modify the manuscript and prefer to submit it to another journal, please notify me of your decision immediately so that the manuscript may be formally withdrawn from consideration by Microbiology Spectrum.

Response to the reviewers:

Reviewer #1

Thank you for your support of our study and the detailed guidance on how to improve the manuscript. Please, find our point-by-point response below.

Introduction - The authors dedicate an entire paragraph to e-cig and periodontal disease. Still, the authors focus on *S. mutans*, which is associated with tooth decay and not with periodontal disease. This seems puzzling to me. Additionally, there does not seem to be any evidence of increased tooth decay due to the use of e-cigs. Thus, based on the introduction, I do not understand why the authors focus on *S. mutans*.

Response: On page 4 of the introduction, we included more rationale to why we focus on Streptococci and references to the literature highlighting the role of *S. mutans* not just in dental caries but also in periodontitis.

Additionally, it is unclear how focusing on three strep species grown as single species will help you better understand the oral microbiome. Maybe you should consider doing multi-species biofilms to see how e-cigs affect these streps when grown under slightly more complex conditions than just pure culture.

Response: While we added more references supporting our rationale to assess the effect of e-cig aerosols on the three *Strep* species we selected (page 4) as well as a new experiment to assess competition between *S. mutans* and the commensals (Figure S1). We show that *S. mutans* outcompetes *S. sanguinis* and *gordonii* (as has been shown by Kreth et al) and using 'conditioned media' from *S. mutans* culture that the suppression of commensal growth is due the secretion of metabolites by *S. mutans* into the media.

We agree with the reviewer that establishing multi-species model communities of higher complexity will be more valuable due to their physiological relevance, and we do plan to do so in future experiments.

Bacterial culture- Minor typo: it should be bacteria were sub-cultured in TSB, not "bacteria was..."

Response: Thank you for catching that!

Growth curve - I am particularly interested in the exposure of TSB to e-cig vapors. The authors mention that TSB was pretreated with e-cig aerosol (10-sec puff, 5-minute exposure). Was this done only once, or were this repeated multiple times? As written, it is unclear to me.

Response: We clarified the preparation of TSB media exposed to e-cig aerosols in the Materials and Methods under E-cigarette aerosol exposure and the following paragraph "Growth Curve". Media were exposed once to a 10-second puff for a 5-minute exposure before every experiment.

Biofilm formation - *sanguinis* and *gordonii* are lousy biofilm formers in the absence of a saliva-coated surface or sugar, like sucrose. To get a little closer to oral conditions, I would encourage the authors to coat their 96-well plates with saliva.

Response: Thank you for your guidance on this! We performed additional biofilm experiments with 1% sucrose supplementation (new Figure S4). While the biofilm for both *S. sanguinis* and *S. gordonii* was enhanced compared to no-sucrose conditions, e-cig aerosol exposure in the presence of sucrose did not enhance biofilm formation for the commensals.

The authors are trying to make the case that a decrease in *S. s* and or *S. g* could favor the growth of *S. m* in the oral cavity. The authors should consider co-culturing the different streps and exposing the multi-species biofilm to e-cig vapor to determine any effect of the growth of the different species.

Response: Following your suggestion, we performed the competition experiments shown in new Fig S4. At this time, we can only show that *S. mutans* outcompetes *S. sanguinis* and *gordonii* (as has been shown by Kreth et al) and that the suppression of commensal growth is due to the secretion of metabolites by *S. mutans* into the media. We have yet to grow multi-species biofilm and expose them to e-cig vapor. We agree with the reviewer this would be a more impactful experiment and we do plan to do so in future experiments.

Page 12 - minor typo. I think the authors may want to say conducive instead of conductive.

Response: Thank you! Corrected!

Page 13 - The authors overstate the link between hydrophobic and host surface attachment, neglecting the fact that interactions with host surfaces can also be mediated by bacterial surface proteins. I encourage the authors to tone their conclusion down here because while hydrophobicity may contribute to surface attachment, it is not the only factor.

Response: We made changes to the text in the hydrophobicity section to highlight the value of the assay in its own without using it as the only read-out for cell attachment capabilities. In addition, we included more factors of changing hydrophobicity in the discussion.

Fig 5B - It is not clear to me what I learned from the experiment shown in fig 5B. There's no quantification (for example, CFU of OKF6 adhered bacteria), nor did the authors investigate the effect of e-cig vapor on bacterial adhesion to OKF6 cells. So what is the point of this experiment?

Response: With the microscopy images of bacteria interacting with oral epithelial cells we aimed to demonstrate binding/attachment and bacterial interaction of *Strep.* species. As can be appreciated from the images the clustered growth of *S. mutans* did not allow for proper quantification. We agree with the reviewer that in the absence of quantification this is not helpful but kept the images to demonstrate the difference in aggregation of *S. mutans* over the other two strains. We performed new attachment

experiments to address the issue of quantification, plated cell lysate and bacteria upon co-culture and included conditions of e-cig exposure as suggested by the reviewer (new Figure 6). *S. mutans* had an overall increased capacity to attach compared to the commensals and appeared to be altered under some conditions of e-cig aerosol exposure but not significantly.

Reviewer #2 (Comments for the Author):

In this study, Catala-Valentin et al. studied the impact of E-cigarette aerosol on the growth and biofilm formation of three oral streptococci. They showed that E-cigarette aerosol inhibited the growth of two commensal *Streptococcus* species, i.e., *S. sanguinis* and *S. gordonii*, but did not affect the growth of an oral pathogen, *S. mutans*. Most of the results shown in this study have been reported by previous publications. In the papers PMID: 31835369 and PMID: 33329035, the authors showed that the growth of oral commensal streptococci were inhibited by E-cigarette aerosol. In PMID: 32683796, E-cigarettes increased the growth and biofilm formation of *S. mutans* on teeth surfaces. More similar publications can be found. Hence, the authors need to show more innovative results.

Response: We appreciate the comments and feedback. We agree that the assessment of e-cig aerosol exposure of *Strep.* species is not novel. However, as we refer to the study findings listed by the reviewer throughout the manuscript, their analysis focused on one species of *Strep.* and could not be compared between each other. More so, as referred to some of these studies had conflicting results in terms of vape exposure modifying growth or with certain e-cig liquids (e.g., flavor, nicotine). Our study included all these conditions and the experiment for all species were done to be directly comparable.

March 2, 2022

Dr. Claudia D. Andl
University of Central Florida
Burnett School of Biomedical Sciences
4111 Libra Dr
Orlando, FL 32816

Re: Spectrum02421-21R1 (E-cigarette aerosol exposure favors the growth and colonization of oral *S. mutans* compared to commensal *Streptococci*)

Dear Dr. Claudia D. Andl:

Your manuscript has been accepted, and I am forwarding it to the ASM Journals Department for publication. You will be notified when your proofs are ready to be viewed.

Sincerely,

Justin Kaspar
Editor, Microbiology Spectrum

Journals Department
Supplemental Figures 1-5: Accept